# Thigh-hip ratio is significantly associated with all-cause mortality among Japanese community-dwelling men

**Ryuichi Kawamoto** [1,2☯]*, **Asuka Kikuchi**[1,2☯], **Daisuke Ninomiya**[1,2☯], **Teru Kumagi**[1☯]

**1** Department of Community Medicine, Ehime University Graduate School of Medicine, Toon, Ehime, Japan,
**2** Department of Internal Medicine, Seiyo Municipal Nomura Hospital, Seiyo, Ehime, Japan

☯ These authors contributed equally to this work.
* rykawamo@m.ehime-u.ac.jp

**Data Availability Statement:** All relevant data are within the paper and its Supporting Information files.

## Abstract

Anthropometric evaluation is a simple yet essential indicator of muscle and fat mass when studying life prognosis in aging. This study aimed to investigate the contributions of anthropometric measurements, independent of body mass index, to measures of all-cause mortality. We examined data for 1,704 participants from the 2014 Nomura Cohort Study who attended follow-ups for the subsequent eight years (follow-up rate: 93.0%). Of these, 765 were male (aged 69 ± 11 years) and 939 were female (aged 69 ± 9 years). The Japanese Basic Resident Registry provided data on adjusted relative hazards for all-cause mortality. The data were subjected to a Cox regression analysis, wherein the time variable was age and the risk factors were gender, age, anthropometric index, smoking habits, drinking habits, exercise habits, cardiovascular history, hypertension, lipid levels, diabetes, renal function, and serum uric acid. Of the total number of participants, 158 (9.3%) were confirmed to have died, and of these, 92 were male (12.0% of all male participants) and 66 were female (7.0% of all female participants). The multivariable Cox regression analysis revealed that a smaller thigh–hip ratio predicted eight-year all-cause mortality in male participants, but only baseline body mass index was associated with all-cause mortality in female participants. Thigh–hip ratio is a useful predictor of death in Japanese community-dwelling men.

## Introduction

The global prevalence of obesity has reached epidemic proportions over the past few decades [1]. Obesity plays a pivotal role in the development of metabolic syndrome, cardiovascular disease (CVD), and morbidity [2]. Clinically, the accumulation of body fat in the abdomen is associated with the highest risk of death due to obesity of any degree [3]. In particular, increased visceral and abdominal fat, independent of overall adiposity, is associated with an increased risk of metabolic diseases [4, 5]. Many studies have shown that anthropometric parameters such as body mass index (BMI) [6], waist circumference [7], hip circumference [8], waist-to-hip ratio [9, 10], and waist-to-height ratio [11–13] serve as useful markers of

**Funding:** No. This work was supported in part by a grant-in-aid from the Foundation for Development of Community (2023). No additional external funding was received for this study. The funders played no role in the study design, data collection and analysis, decision to publish, or manuscript preparation.

**Competing interests:** The authors have no competing interests to declare.

metabolic syndrome, CVD, and all-cause mortality. These markers also indicate visceral adiposity, a critical factor influencing obesity-related mortality.

Compared with upper-body fat, peripheral fat in the lower body has contrasting (i.e., both beneficial and detrimental) associations with long-term blood pressure, subclinical atherosclerosis, and diabetes [14, 15]. Thigh circumference reflects quadriceps muscle mass and peripheral subcutaneous fat [16] and has the opposite associations to upper-body fat with risk of insulin resistance, atherosclerosis, type 2 diabetes [13, 17], CVD mortality, and all-cause mortality [18]. However, the relationship between thigh circumference and risk of all-cause mortality in community-dwelling people is under-investigated [18, 19]. There is also limited research on the association between anthropometric indices and all-cause mortality.

Thus, using cohort data for Japanese community-dwelling people, we aimed to investigate whether waist, thigh, and hip circumference and their ratios are associated with risk for all-cause mortality.

## Materials & methods

### Study design and participants

We designed the study as a prospective cohort study that was a component of the Nomura study [20], and all procedures were performed in compliance with the related standards and laws. The Nomura Health and Welfare Center is in a remote location of Japan's Ehime Prefecture. The study commenced in 2014 and was conducted on individuals from the region. A total of 1,832 individuals, ranging in age at enrollment from 22 to 95 years, participated in a community-based annual health check. All participants completed a self-administered questionnaire on physical activity, medical history, present health condition, and prescribed medication use (e.g., antihypertensive, antidyslipidemic, antidiabetic, and uric-acid [UA] lowering drugs). The flowchart for participant enrollment and exclusion is shown in Fig 1. Following the baseline examination, 1,717 participants were tracked for eight years, and the Japanese Basic Resident Registry database was used to confirm whether they were still alive or had passed away. The study protocol was examined and approved by the Ehime University Hospital's Institutional Review Board (IRB: 1402009). All participants gave their written informed consent.

### Evaluation of risk factors

The participants' current status, exercise habits, details about their medical history, and medications were collected through a structured questionnaire interview. To qualify as having exercise habits, the participants must have participated in any type of moderate-to-vigorous physical activity, such as brisk walking, golfing, gardening, jogging, or playing tennis, for at least 30 minutes on at least 2 days per week [$\geq$ 600 metabolic equivalent tasks (MET)-minutes/week] for a minimum of 1 year. All anthropometric measurements were in accordance with the WHO standards [21]. Body weight was measured to the nearest 0.1 kg with an electronic standard weight scale machine (HBF-214, Omron, Tokyo, Japan), with participants wearing underwear or light clothing and body height was measured without shoes to the nearest 1 cm using a stadiometer (DSN-90, Muratec-KDS, Kyoto, Japan). Body mass index (BMI) was determined as weight (kg) divided by squared height ($m^2$). Waist, hip, and thigh circumferences (cm) were measured with an anthropometric tape (25–204, Clover, Osaka, Japan). Hip circumference was measured at the widest level over the greater trochanters, and waist circumference was estimated in the horizontal plane at the midpoint between the anterior iliac crest and the lower edge of the ribs. Thigh circumference was measured just below the gluteal fold on both legs and the mean of the two was used in the analysis. Waist-to-hip (waist–hip)

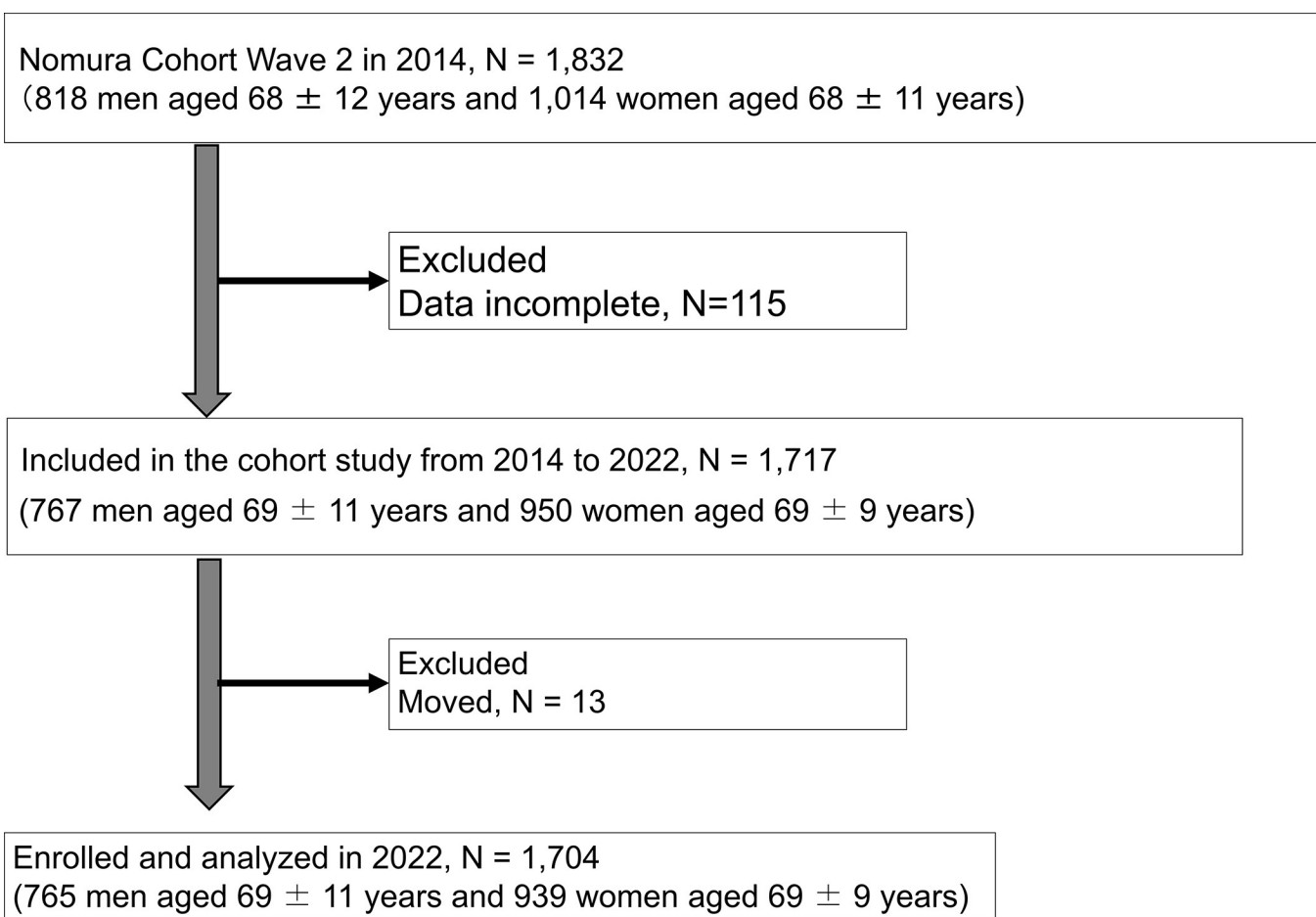

Nomura Cohort Wave 2 in 2014, N = 1,832
（818 men aged 68 ± 12 years and 1,014 women aged 68 ± 11 years）

Excluded
Data incomplete, N=115

Included in the cohort study from 2014 to 2022, N = 1,717
(767 men aged 69 ± 11 years and 950 women aged 69 ± 9 years)

Excluded
Moved, N = 13

Enrolled and analyzed in 2022, N = 1,704
(765 men aged 69 ± 11 years and 939 women aged 69 ± 9 years)

**Fig 1. Flowchart of participants.**

ratio by dividing waist circumference by hip circumference, and thigh-to-waist (thigh–waist) ratio was computed by dividing thigh circumference by waist circumference. Similarly, the thigh to hip (thigh–hip) ratio was calculated by dividing thigh circumference by hip circumference. Smoking habits (pack-years) were estimated by multiplying the number of years the person had been a smoker by the typical number of packs consumed per day. Accordingly, subjects were divided into three categories: former smokers, light smokers ($< 20$ pack-years), and heavy smokers ($> 20$ pack-years). The amount of alcohol consumed each day was calculated using Japanese alcoholic beverage units (22.9 g ethanol), and participants were divided into four groups: nondrinkers, occasional drinkers ($< 1$ unit/day), moderate daily drinkers (1–2 units/day), and heavy daily drinkers (3 units/day); none drank more than 3 units/day [22]. Systolic and diastolic blood pressure (SBP and DBP, respectively) were measured using an automated sphygmomanometer (BP-103i; Colin, Aichi, Japan) with an appropriately sized cuff placed on their right upper arm while the subjects were seated after at least 5 minutes of immobility, and the average of two consecutive measurements was used for calculation. Participants were asked to fast overnight before measurements and their triglycerides (TG), high-density lipoprotein cholesterol (HDL-C), low-density lipoprotein cholesterol (LDL-C), serum uric acid (SUA), and creatinine (Cr) (enzymatic assay, Kyowa Medex, Tokyo, Japan) were measured using an automated analyzer (Hitachi, Tokyo, Japan). Hemoglobin A1c (HbA1c) was analyzed by the latex agglutination inhibition method (Kyowa Medex, Tokyo, Japan)

using an automated analyzer (DM-JACK® Ex, Tokyo, Japan). The glomerular filtration ratio (eGFR) was estimated using the Chronic Kidney Disease Epidemiology Collaboration (CKD-EPI) formula with the Japanese coefficients: $141 \times (Cr/0.9)^{-0.411} \times 0.993^{age} \times 0.813$ for male participants with Cr levels < 0.9 mg/dL and $141 \times (Cr/0.9)^{-1.209} \times 0.993^{age} \times 0.813$ for male participants with Cr levels > 0.9 mg/dL. The equation was $144 \times (Cr/0.7)^{-0.329} \times 0.993^{age} \times 0.813$ for females with Cr levels < 0.7 mg/dL and $144 \times (Cr/0.7)^{-1.209} \times 0.993^{age} \times 0.813$ for females with Cr levels > 0.7 [23].

Participants were deemed to have hypertension if their SBP was > 140 mmHg, DBP was > 90 mmHg, and/or they were on antihypertensive medications. They were considered to have high LDL cholesterol if their LDL-C level was > 140 mg/dL and/or if they were on antidyslipidemic medication, low HDL cholesterolemia if their HDL-C level was < 40 mg/dL for males and < 50 mg/dL for females, and hypertriglyceridemia if their TG level was ≥ 150 mg/dL. Participants were considered diabetic if their HbA1c was ≥ 6.5% and/or they were on antidiabetic medication. They were classified as hyperuricemia if their SUA was ≥ 7.0 mg/dL and/or they were on SUA-reducing medication. They were diagnosed with CKD based on an eGFR of < 60 mL/min/1.73 $m^2$ or the presence of proteinuria. Cardiovascular disease includes self-reported ischemic heart disease, stroke, and peripheral vascular disease.

## Statistical analysis

The statistical analysis was performed using IBM SPSS Statistics 26.0 (SPSS, Armonk, NY, USA). If the data were normally distributed, we used the mean and standard deviation (SD) to express continuous variables; if not, we used median (interquartile range). For parameters with non-normal distributions, we used log-transformed values in all analyses. Categorical and continuous data were compared by conducting chi-squared analyses of categorical variables and student's *t*-tests on normally distributed continuous variables. Predictors of all-cause mortality were determined by calculating the area under the receiver operating characteristic (ROC) curve for each variable. The ROC curve plots sensitivity (true positive rate) versus 1 − specificity (false positive rate) for each marker evaluated. The gold standard ROC curve is the line connecting (0,0) to (0,1) and (0,1) to (1,1). In general, the ROC curve lies between these two extremes; the area under the ROC curve is essentially a summary measure that averages diagnostic accuracy across the spectrum of test values. A diagonal line with an area under the curve (AUR) close to 0.5 represents a nondiagnostic indicator. We next conducted a univariate analysis using the Cox proportional hazards model for the following baseline anthropometric variables: thigh circumference, age, BMI, smoking habits, drinking habits, exercise habits, history of CVD, hypertension, hypertriglyceridemia, low HDL-cholesterolemia, high LDL-cholesterolemia, diabetes, CKD, and hyperuricemia. The multivariable analysis was based on the Cox proportional hazards model and was conducted using the forced entry method. Subgroup analyses were performed to assess the consistency of the observed correlation between thigh–hip ratio and all-cause mortality. Age (< 65 years or ≥ 65 years), BMI (< 25 kg/$m^2$ or ≥ 25 kg/$m^2$ [24]), exercise habits (yes or no), and time to death (< 750 days or ≥ 750 days) were all analyzed. Except for the effect variable, all confounding variables that were significant in the univariate analysis were corrected for when performing the interaction test on the effect variable. Differences with *p*-values < 0.05 were considered statistically significant.

## Results

### Baseline characteristics of participants by gender and survival status

The study participants were 765 males aged 69 ± 11 years (range: 24–90 years) and 939 females aged 69 ± 9 years (range: 26–90 years) (Table 1). The proportions of participants with CVD,

**Table 1. Baseline characteristics of participants by gender and survival status.**

| Baseline characteristics (N = 1,704) | Men (N = 765) | | | Women (N = 939) | | |
|---|---|---|---|---|---|---|
| | Alive (N = 673) | Dead (N = 92) | p | Alive (N = 873) | Dead (N = 66) | p |
| Age (years) | 68 ± 11 | 76 ± 9 | **< 0.001** | 69 ± 9 | 75 ± 7 | **< 0.001** |
| Smoking habits (non = 1, ex = 2, light = 3, heavy = 4), % | 41.6/37.9/6.4/14.1 | 41.3/44.6/5.4/8.7 | 0.421 | 96.3/2.4/0.8/0.5 | 98.5/0/1.5/0 | 0.515 |
| Drinking habits (non = 1, occasional = 2, light = 3, heavy = 4), % | 23.3/23.2/15.0/38.5 | 29.3/21.7/21.7/27.2 | 0.094 | 70.8/22.3/4.5/2.4 | 74.2/22.7/3.0/0 | 0.576 |
| Exercise habits, n (%) | 245 (36.4) | 33 (35.9) | 1.000 | 339 (38.8) | 18 (27.3) | 0.066 |
| History of cardiovascular disease, n (%) | 61 (9.1) | 13 (14.1) | 0.132 | 33 (3.8) | 5 (7.6) | 0.180 |
| Hypertension, n (%) | 432 (64.2) | 66 (71.7) | 0.163 | 535 (61.3) | 51 (77.3) | **0.012** |
| Systolic blood pressure (mmHg) | 135 ± 17 | 140 ± 19 | **0.007** | 135 ± 18 | 142 ± 16 | **0.004** |
| Diastolic blood pressure (mmHg) | 80 ± 10 | 80 ± 10 | 0.280 | 76 ± 10 | 79 ± 9 | 0.067 |
| Antihypertensive medication, n (%) | 297 (44.1) | 42 (45.7) | 0.823 | 370 (42.4) | 32 (48.5) | 0.367 |
| Hypertriglyceridemia, n (%) | 121 (18.0) | 16 (17.4) | 1.000 | 107 (12.3) | 8 (12.1) | 1.000 |
| Triglycerides (mg/dL) | 92 (68–131) | 85 (69–137) | 0.742 | 86 (65–117) | 88 (65–117) | 0.978 |
| Low HDL-cholesterolemia, n (%) | 46 (6.8) | 10 (10.9) | 0.196 | 108 (12.4) | 8 (12.1) | 1.000 |
| HDL cholesterol (mg/dL) | 62 ± 16 | 59 ± 18 | 0.107 | 69 ± 17 | 66 ± 14 | 0.253 |
| High LDL-cholesterolemia, n (%) | 202 (30.0) | 31 (33.7) | 0.471 | 475 (54.4) | 35 (53.0) | 0.898 |
| LDL cholesterol (mg/dL) | 114 ± 28 | 107 ± 32 | **0.017** | 125 ± 29 | 120 ± 29 | 0.155 |
| antidyslipidemic medication, n (%) | 86 (12.8) | 17 (18.5) | 0.143 | 246 (28.2) | 21 (31.8) | 0.572 |
| Diabetes, n (%) | 112 (16.6) | 17 (18.5) | 0.657 | 75 (8.6) | 6 (9.1) | 0.821 |
| Hemoglobin A1c (%) | 5.7 (5.4–6.0) | 5.7 (5.3–6.1) | 0.899 | 5.7 (5.5–5.9) | 5.7 (5.5–6.0) | 0.274 |
| Antidiabetic medication, n (%) | 87 (12.9) | 10 (10.9) | 0.738 | 44 (5.0) | 5 (7.6) | 0.382 |
| Chronic kidney disease, n (%) | 153 (22.7) | 39 (42.4) | **< 0.001** | 133 (15.2) | 18 (27.3) | **0.015** |
| eGFR (mL/min/1.73 m$^2$) | 72.1 ± 11.7 | 62.9 ± 16.7 | **< 0.001** | 73.3 ± 10.9 | 67.1 ± 14.2 | **< 0.001** |
| Proteinuria, n (%) | 83 (12.3) | 21 (23.1) | **0.008** | 60 (6.9) | 12 (18.2) | **0.003** |
| Hyperuricemia, n (%) | 186 (27.6) | 22 (23.9) | 0.532 | 116 (13.3) | 13 (19.7) | 0.141 |
| SUA (mg/dL) | 6.0 ± 1.3 | 5.9 ± 1.5 | 0.385 | 4.7 ± 1.1 | 4.6 ± 1.3 | 0.369 |
| UA-lowering medication, n (%) | 54 (8.0) | 5 (5.4) | 0.531 | 7 (0.8) | 1 (1.5) | 0.443 |

HDL, high-density lipoprotein; LDL, low-density lipoprotein; eGFR, estimated glomerular filtration ratio; SUA, serum uric acid; UA, serum uric acid.

Data presented are means ± standard deviation. Data for triglycerides and hemoglobin A1c were skewed and are presented as median (interquartile range) values, and they were log-transformed for analysis.

P-values are from Student's t-tests for continuous variables or $\chi^2$ tests for categorical variables.

Significant values ($p < 0.05$) are presented in bold.

hypertension, hypertriglyceridemia, low-HDL cholesterolemia, high LDL-cholesterolemia, diabetes, hyperuricemia, and CKD were 6.6%, 63.6%, 14.8%, 10.1%, 43.6%, 12.3%, 20.1%, and 19.8%, respectively. The follow-up study reported 158 deaths (92 males and 66 females), giving a mortality rate of 5,353,968 person-days (average follow-up period: 2,942 days). Relative to the participants who were still living, those who died had a significantly higher baseline age, SBP, and prevalence of CKD in both gender groups.

## Baseline anthropometric indices by gender and survival status

Compared with the participants who were still living, the deceased participants had a significantly smaller thigh circumference and thigh–hip ratio (Table 2). This finding was true for both gender groups. Males who died had a significantly smaller thigh–waist ratio and females had a significantly smaller hip circumference. However, their BMI and waist circumference did not differ significantly.

**Table 2. Baseline anthropometric indices for participants by gender and survival status.**

| Baseline characteristics (N = 1,717) | Men (N = 765) | | | Women (N = 939) | | |
|---|---|---|---|---|---|---|
| | Alive (N = 673) | Dead (N = 92) | p | Alive (N = 873) | Dead (N = 66) | p |
| Body mass index (kg/m²) | 23.2 ± 2.9 | 23.1 ± 3.7 | 0.676 | 22.6 ± 3.2 | 22.0 ± 3.6 | 0.164 |
| Waist circumference (cm) | 82.4 ± 7.9 | 82.4 ± 10.3 | 0.986 | 80.5 ± 9.0 | 78.5 ± 10.2 | 0.079 |
| Hip circumference (cm) | 91.2 ± 5.1 | 91.2 ± 6.7 | 0.972 | 90.2 ± 5.8 | 88.5 ± 6.5 | **0.024** |
| Thigh circumference (cm) | 44.4 ± 3.8 | 42.5 ± 4.4 | **< 0.001** | 44.0 ± 3.9 | 41.8 ± 4.2 | **< 0.001** |
| Waist–hip ratio | 0.90 ± 0.06 | 0.90 ± 0.07 | 0.923 | 0.89 ± 0.06 | 0.88 ± 0.07 | 0.398 |
| Thigh–waist ratio | 0.54 ± 0.04 | 0.52 ± 0.05 | **< 0.001** | 0.55 ± 0.06 | 0.54 ± 0.06 | 0.074 |
| Thigh–hip ratio | 0.49 ± 0.03 | 0.47 ± 0.03 | **< 0.001** | 0.49 ± 0.03 | 0.47 ± 0.03 | **< 0.001** |

$P$-values are from Student's $t$-tests for continuous variables.

Significant values ($p < 0.05$) are presented in bold.

## Area under the ROC curve for all-cause mortality according to baseline anthropometric indices

Table 3 and Fig 2 present the gender-specific area under the ROC curve (AUC) for individual anthropometric indices used to predict all-cause mortality. Thigh–hip ratio yielded the lowest AUC for males (0.311, 95% confidence interval [CI]: 0.251–0.371), and thigh circumference yielded the lowest AUC for females (0.356, 95% CI: 0.283–0.430).

## Kaplan–Meier survival curves for four categories of thigh–hip ratio

Kaplan–Meier survival curves were estimated for four categories of thigh–hip ratio to determine patterns in the relationships between this measure and all-cause mortality (Fig 3). The measure was divided into categories using intervals approximating one SD of the sex difference in thigh–hip ratio to ensure that stable risk estimates were obtained for both gender groups. The results indicated that the association patterns were similar for the thigh–hip ratio for both genders, and mortality risk was significantly greater for individuals in the smallest thigh–hip ratio category than for those in all other categories ($p < 0.001$ for both genders).

## Hazard ratio (95% CI) for all-cause mortality by baseline anthropometric indices

Thigh circumference (HR: 0.63; 95% CI: 0.50–0.79), thigh–waist ratio (HR: 0.77; 95% CI: 0.60–0.99), and thigh–hip ratio (HR: 0.64; 95% CI: 0.49–0.84) were significant predictors for

**Table 3. Area under the curve for all-cause mortality of baseline anthropometric indexes.**

| Baseline characteristics (N = 1,704) | Men (N = 765) | | Women (N = 939) | |
|---|---|---|---|---|
| | AUC (95% CI) | p | AUR (95% CI) | p |
| Body mass index | 0.479 (0.403–0.547) | 0.523 | 0.441 (0.366–0.516) | 0.109 |
| Waist circumference | 0.496 (0.428–0.564) | 0.897 | 0.434 (0.359–0.509) | 0.074 |
| Hip circumference | 0.480 (0.413–0.547) | 0.527 | **0.413 (0.339–0.488)** | **0.019** |
| Thigh circumference | **0.378 (0.312–0.444)** | **< 0.001** | **0.356 (0.283–0.430)** | **< 0.001** |
| Waist–hip ratio | 0.508 (0.438–0.577) | 0.813 | 0.471 (0.397–0.545) | 0.433 |
| Thigh–waist ratio | 0.358 (0.297–0.419) | **< 0.001** | 0.431 (0.362–0.501) | 0.062 |
| Thigh–hip ratio | **0.311 (0.251–0.371)** | **< 0.001** | **0.386 (0.314–0.458)** | **0.002** |

AUC, area under the curve; CI, confidence interval.

Significant values ($p < 0.05$) are presented in bold.

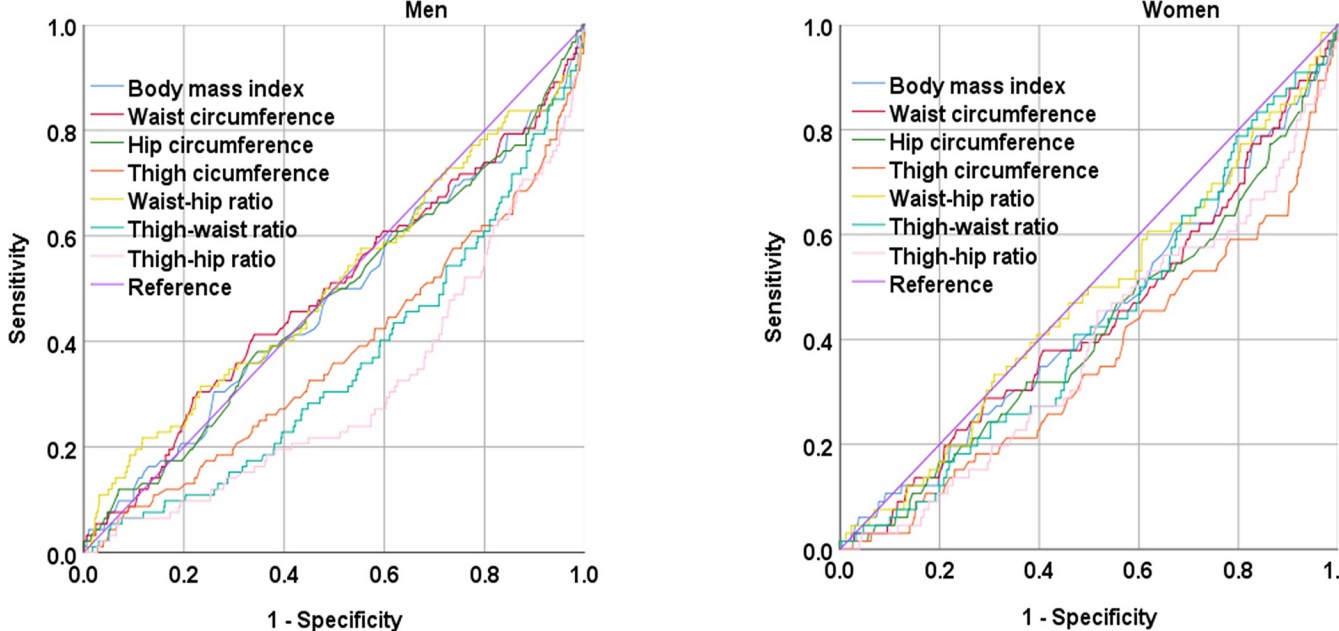

**Fig 2. Receiver operating characteristic curve analysis for all-cause mortality according to baseline anthropometric indices by gender.** Thigh–hip ratio yielded the lowest area under the curve (AUC) for male participants, and thigh circumference yielded the lowest AUC for female participants.

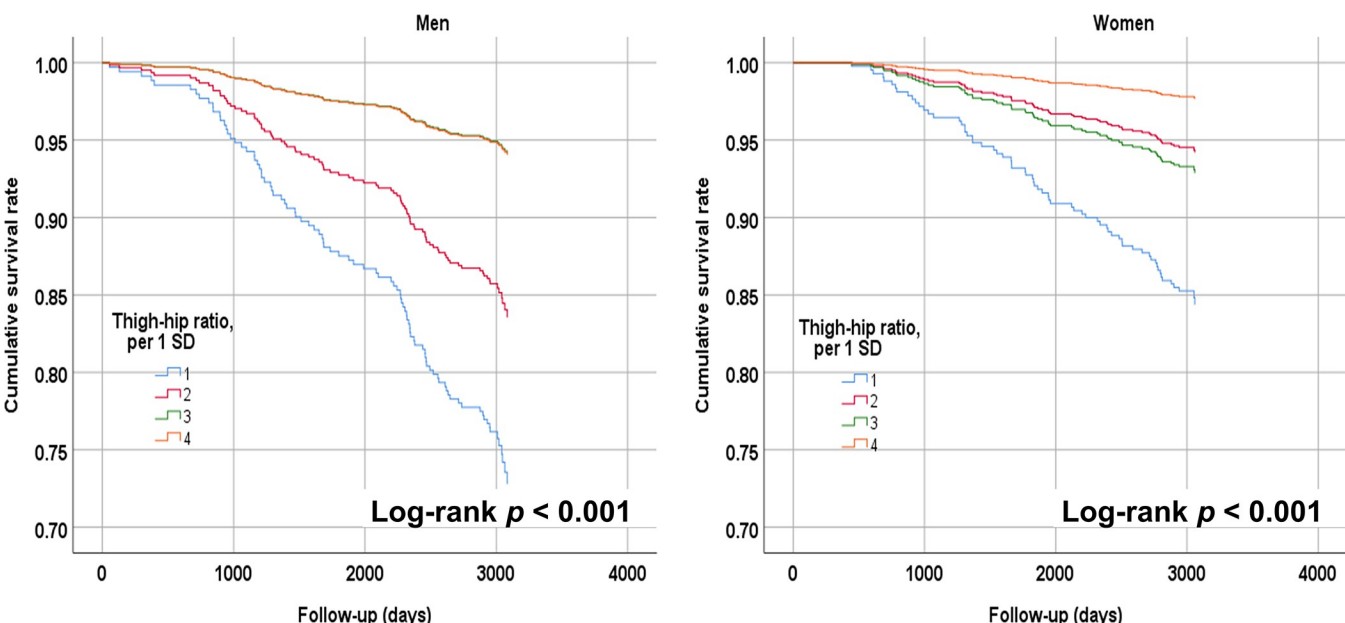

**Fig 3. Kaplan–Meier survival curves for four categories of thigh–hip ratio by gender.** Thigh–hip ratio showed similar association patterns for both gender groups, and mortality risk was significantly greater for individuals with the smallest category of thigh–hip ratio than for those in all other categories ($p < 0.001$ for both genders).

**Table 4. Hazard ratio (95% CI) for all-cause mortality by baseline anthropometric indexes.**

| | Men (N = 765) | | | | Women (N = 939) | | | |
|---|---|---|---|---|---|---|---|---|
| | Univariable | | Multivariable | | Univariable | | Multivariable | |
| Baseline characteristics | HR (95% CI) | p | HR (95% CI) | p | HR (95% CI) | p | HR (95% CI) | p |
| Body mass index, per 1 SD kg/m$^2$ | 0.93 (0.75–1.17) | 0.553 | 1.00 (0.78–1.27) | 0.968 | 0.81 (0.62–1.05) | 0.117 | **0.74 (0.56–0.99)** | **0.044** |
| Waist circumference, per SD cm | 0.93 (0.75–1.17) | 0.552 | 0.86 (0.58–1.29) | 0.474 | 0.82 (0.63–1.06) | 0.135 | 0.94 (0.62–1.42) | 0.755 |
| Hip circumference, per 1 SD cm | 0.88 (0.70–1.11) | 0.283 | 0.95 (0.66–1.36) | 0.768 | **0.72 (0.55–0.94)** | **0.017** | 0.82 (0.53–1.25) | 0.349 |
| Thigh circumference, per 1 SD cm | **0.63 (0.50–0.79)** | **< 0.001** | **0.70 (0.50–0.99)** | **0.042** | **0.57 (0.43–0.76)** | **< 0.001** | 0.82 (0.56–1.22) | 0.326 |
| Waist–hip ratio, per 1 SD unit | 1.06 (0.85–1.32) | 0.620 | 1.00 (0.76–1.32) | 0.999 | 0.91 (0.71–1.18) | 0.481 | 0.98 (0.73–1.31) | 0.877 |
| Thigh–waist ratio, per 1 SD unit | **0.61 (0.49–0.77)** | **< 0.001** | **0.77 (0.60–0.99)** | **0.047** | 0.84 (0.65–1.09) | 0.197 | 1.00 (0.75–1.33) | 0.991 |
| Thigh–hip ratio, per 1 SD unit | **0.51 (0.40–0.64)** | **< 0.001** | **0.64 (0.49–0.84)** | **0.001** | **0.62 (0.47–0.82)** | **0.001** | 0.90 (0.66–1.23) | 0.509 |

HR, hazard ratio; CI, confidence interval; SD, standard deviation. Each anthropometric measure was added individually to the model.

The multivariable model was adjusted for age, body mass index, smoking status, drinking status, exercise habits, history of cardiovascular disease, hypertension, low HDL-cholesterolemia, high LDL-cholesterolemia, hypertriglyceridemia, diabetes, chronic kidney disease, hyperuricemia at baseline.

Significant values (*p* < 0.05) are presented in bold.

all-cause mortality in males (Table 4). In contrast, only BMI was a significant predictor for all-cause mortality in females (HR: 0.74; 95% CI: 0.56–0.99).

## Hazard ratio (95% CI) for all-cause mortality by baseline thigh–hip ratio in males

We conducted sub-analyses to analyze the adjusted HRs of baseline thigh–hip ratio for all-cause mortality in males stratified based on age (< 65 years, ≥ 65 years), BMI (< 25 kg/m$^2$, ≥ 25 kg/m$^2$), exercise habits (yes, no), and time till death (< 730 days, ≥ 730 days) (Table 5). Similar to our previous results, a smaller thigh–hip ratio was associated with a greater risk of all-cause mortality, and this relationship was particularly pronounced among participants with BMI < 25 kg/m$^2$ and who did not have exercise habits. In addition, there was a significant interaction between BMI and thigh–hip ratio.

## Discussion

A key conclusion of this cohort study is that the anthropometric measure of thigh–hip ratio is a substantial and independent predictor of all-cause mortality in community-dwelling men, even after adjustment for age, BMI, smoking status, drinking status, exercise habits, history of CVD, hypertension, low HDL-cholesterolemia, high LDL-cholesterolemia, hypertriglyceridemia, diabetes, CKD, hyperuricemia at baseline. Smaller thigh–hip ratio was associated with death, however, only in individuals with BMI < 25 kg/m$^2$. The study also highlighted an interaction between thigh–hip ratio and BMI in the context of all-cause mortality. To the best of our knowledge, this study is the first to demonstrate an association between thigh–hip ratio and all-cause mortality in community-dwelling men in Japan. However, it must be considered that some important confounding factors were not identified or adjusted for.

Despite the relationship between thigh circumference and all-cause mortality having received substantial attention, no study exists on the relationship between thigh–hip ratio and all-cause mortality. The size of the thigh is frequently used as a measure of muscle and peripheral subcutaneous fat in the body and has been linked to both insulin resistance, risk of diabetes, and atherosclerosis [25, 26]. The Danish MONICA (monitoring trends in and determinants of cardiovascular disease) project, involving 1,436 male and 1,380 female participants, reported inverse correlations between thigh circumference and all-cause mortality as

**Table 5. Hazard ratio (95% CI) for all-cause mortality by baseline thigh–hip ratio in men by sub-analysis.**

| | Men Thigh–hip ratio (N = 765) | | | |
|---|---|---|---|---|
| | N | HR (95% CI) | P | P for interaction |
| Age | | | | |
| < 65 years | 215 | 0.66 (0.29–1.47) | 0.308 | 0.075 |
| ≥ 65 years | 550 | **0.64 (0.48–0.86)** | **0.003** | |
| BMI | | | | |
| < 25 kg/m$^2$ | 579 | **0.57 (0.41–0.79)** | **0.001** | **0.013** |
| ≥ 25 kg/m$^2$ | 186 | 0.87 (0.50–1.49) | 0.604 | |
| Exercise habits | | | | |
| No | 487 | **0.65 (0.47–0.91)** | **0.011** | 0.931 |
| Yes | 278 | 0.66 (0.40–1.07) | 0.092 | |
| Time to death | | | | |
| < 730 days | 7 | ------ | ------ | ------ |
| ≥ 730 days | 758 | **0.62 (0.47–0.82)** | **0.001** | |

HR, hazard ratio; CI, confidence interval.

Multivariable-adjusted for age, body mass index, smoking status, drinking status, exercise habits, history of cardiovascular disease, hypertension, low HDL-cholesterolemia, high LDL-cholesterolemia, hypertriglyceridemia, diabetes, chronic kidney disease, hyperuricemia at baseline.

Each anthropometric measure was added individually to the model.

Significant values ($p < 0.05$) are presented in bold.

well as morbidity from CVD for both gender groups [19]. A retrospective cohort study on 11,871 patients in the 1999–2004 National Health and Nutrition Examination Survey (NHANES) showed a statistically significant association between greater thigh circumference and lower risk of all-cause and CVD mortality [27]. In a six-year follow-up component of the Nomura Cohort Study, comprising 787 male (aged 69 ± 11 years) and 963 female (aged 69 ± 9 years) participants, thigh circumference and handgrip strength were shown to be useful predictors of death [28]. The present study showed that smaller thigh circumference and thigh–hip ratio are significantly associated with an increased risk of all-cause mortality in men. In other words, male health and survival were negatively impacted when their thigh-hip ratio was smaller, especially if they had a relatively low body weight (BMI < 25 kg/m$^2$). Further, obesity was associated with thigh–hip ratio in males. We found no association between thigh–hip ratio and all-cause mortality in overweight males (BMI ≥ 25 kg/m$^2$), whereas a significant positive correlation existed for participants whose BMI was < 25 kg/m$^2$. Consequently, people with a thigh–hip ratio greater than the 50th percentile and a BMI ≥ 25 kg/m$^2$ may not be at an increased risk of all-cause mortality. Using data from the 1999–2006 NHANES, Chen et al. [18] discovered an additional independent and inverse relationship among adult thigh circumference, all-cause mortality, and cardiovascular death. They found that BMI was a significant effect modifier among individuals who had a BMI < 25 kg/m$^2$ ($p < 0.0001$).

The correlation between thigh circumference and overall mortality remains to be fully understood. Thigh circumference is often used as an indicator of body fat distribution and muscle mass. One study showed that postmenopausal women with normal BMI are at an increased risk of CVD when they have increased trunk fat and reduced leg fat [29]. Several studies have reported that low levels of thigh subcutaneous fat are correlated with poor lipid and glucose metabolism, independent of high abdominal fat, whereas high levels of thigh subcutaneous fat are associated with favorable lipid and glucose metabolism [30]. Research has demonstrated that lower-body muscle size is strongly associated with the development of type 2 diabetes, since insulin resistance is thought to originate in the lower-body muscles (e.g., leg

muscles) and not in the arm muscles [31]. This effect can be attributed to the anti-atherogenic properties of peripheral fat. In addition, skeletal muscle is an endocrine organ and releases numerous cytokines and peptides (also known as myokines) into the blood, which contribute to reducing inflammation [32].

Previous studies have indicated that smaller-than-expected hip circumferences are related to reduced femoral fat, tiny pelvic bone structure, or muscle atrophy in the gluteofemoral region [33]. Conversely, larger hip and thigh circumferences are associated with more favorable lipid and glucose levels [8, 34, 35]. When not accounting for hip circumference, the impact of central obesity on mortality risk is grossly underestimated. A Framingham-type model for CVD mortality showed a significant boost in prediction power when waist and hip circumference were included [36]. While, the People's Republic of China Study on 1,144 male and 1,776 female participants reported that a larger hip circumference was not associated with reduced incident risk factors in male participants [37]. Our study also did not link hip circumference with all-cause mortality. Previous studies have indicated that gluteal subcutaneous fat and muscle mass are assessed through hip circumference [38], while waist circumference primarily indicates subcutaneous and visceral adipose tissue levels. Interestingly, hip circumference shows a strong correlation with BMI, and there is also a correlation between BMI and waist circumference [39]. Hence, a larger hip circumference could suggest higher BMI and waist circumference. To obtain a more accurate estimation of the positive effects of femoral muscle mass, it might be beneficial to adjust thigh circumference (which reflects beneficial fat and muscle mass) in relation to hip circumference or waist circumference (which includes harmful visceral fat). Furthermore, this approach may more accurately assess the benefit of thigh muscle mass for male participants with lower BMIs. These sex-specific differences imply that the biological processes causing these disparities may be influenced by genes, sex hormones, and gender-based differences in hip size, body composition, and/or fat distribution [37]. However, the factors contributing to this relationship remain unknown. Thus, further research is required to understand the underlying biological mechanisms behind these associations.

This study has several strengths. Its prospective design, made possible by the extensive study period and follow-up analysis, makes a significant addition to the literature on this topic. Additional benefits are the measurement of anthropometric indices with adjustments for various potential confounding variables and the inclusion of a sensitivity analysis. However, this study also has some limitations. First, the individuals were mainly healthy middle-aged and older adults who participated in health examinations and lived in a rural area of Japan with a rapidly aging population. Second, we used the all-cause mortality outcome of the Japanese Basic Resident Registry as our benchmark. The possibility of follow-up was limited because the registry does not contain information for participants who relocated during the study period. Third, we did not account for confounding variables, drugs, underlying conditions, and lifestyle changes at both baseline and throughout the follow-up period, which could affect results. Anthropometric measurements and health status can change significantly over time, particularly in an ageing population. Collecting first and repeated information regarding these aspects could offer valuable insights for a more comprehensive understanding of the connection between anthropometric parameters and mortality. Fourth, there was a lack of direct and detail data concerning exercise habits, muscle mass, and body composition which are important for maintaining physique and health. Fifth, in Japan, where the prevalence and severity of obesity is still mild, we examined the interaction between BMI and thigh–hip ratio on mortality in two groups based on a BMI of $\geq 25$ kg/m$^2$. However, the results are not necessarily generalizable. Finally, the thigh–hip ratio, BMI, and all-cause mortality causal association may

have been overestimated, given the small number of participants and fatalities. To clearly explain this, a clear mechanism needs to be found, which will require additional study.

## Conclusions

The present conclusions are based on an eight-year follow-up study of adults ($\geq$ 20 years). They demonstrate that, after correcting for potential confounding variables such as age, body mass index, smoking status, drinking status, exercise habits, history of CVD, hypertension, low HDL-cholesterolemia, high LDL-cholesterolemia, hypertriglyceridemia, diabetes, CKD, and hyperuricemia at baseline, a smaller thigh–hip ratio in males and a lower BMI in females predict increased all-cause mortality.

## Supporting information

**S1 Checklist. STROBE statement—checklist of items that should be included in reports of observational studies.**
(DOCX)

**S1 Data. The minimal anonymized data set necessary to replicate our study findings.**
(XLSX)

## Acknowledgments

We thank Uni-edit (https://uni-edit.net/) for editing and proofreading this manuscript.

## Author Contributions

**Conceptualization:** Ryuichi Kawamoto.

**Data curation:** Ryuichi Kawamoto, Asuka Kikuchi, Daisuke Ninomiya, Teru Kumagi.

**Formal analysis:** Ryuichi Kawamoto.

**Funding acquisition:** Ryuichi Kawamoto.

**Investigation:** Ryuichi Kawamoto.

**Methodology:** Ryuichi Kawamoto.

**Project administration:** Ryuichi Kawamoto.

**Resources:** Ryuichi Kawamoto.

**Supervision:** Teru Kumagi.

**Writing – original draft:** Ryuichi Kawamoto.

**Writing – review & editing:** Ryuichi Kawamoto.

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
