## [Decision Letter · Decision Letter 0]

21 Jul 2023

PONE-D-23-14105Thigh-hip ratio is significantly associated with all-cause mortality among Japanese community-dwelling menPLOS ONE

Dear Dr. Kawamoto,

Thank you for submitting your manuscript to PLOS ONE. After careful consideration, we feel that it has merit but does not fully meet PLOS ONE’s publication criteria as it currently stands. Therefore, we invite you to submit a revised version of the manuscript that addresses the points raised during the review process.

Please clearly address the contradiction between the Kaplan-Meier results and your general conclusionThis manuscript requires a major revisionPlease address all the comments from all the reviewers ==============================

We look forward to receiving your revised manuscript.

Kind regards,

Fredirick Lazaro mashili, MD, PhD

Academic Editor

PLOS ONE

“We express our gratitude to Uni-edit (https://uni-edit.net/) for their valuable assistance in editing and proofreading this manuscript. This research received partial financial support from a grant-in-aid provided by the Foundation for Development of Community (2023). It is important to note that the funders played no part in the study design, data collection and analysis, decision to publish, or manuscript preparation.”

“No.  This work was supported in part by a grant-in-aid from the Foundation for Development of Community (2023). No additional external funding was received for this study. The funders played no role in the study design, data collection and analysis, decision to publish, or manuscript preparation.”

7. Please include your tables as part of your main manuscript and remove the individual files. Please note that supplementary tables (should remain/ be uploaded) as separate "supporting information" files

Additional Editor Comments:

In this study the authors aimed to investigate the contributions of anthropometric measurements, independent of body mass index, to measures of all-cause mortality. They examined data for 1,704 participants from the 2014 Nomura Cohort Study who attended follow-ups for the subsequent eight years (follow-up rate: 93.0%).

This paper presents an intriguing study into the potential association of thigh-hip ratio with all-cause mortality in a Japanese population. There are, however, several areas that can benefit from more precise explanation, and there are certain limitations in the research that need to be addressed.

Study Design and Execution: The research is commendable for its prospective design, the extensive duration of the follow-up period, and the inclusion of sensitivity analyses and adjustment for confounding variables. The use of different anthropometric indices and their inclusion in the analysis provides a multi-dimensional perspective. The Kaplan-Meier survival curves were also useful tools in understanding the relationship between thigh-hip ratio and all-cause mortality. While the authors adjusted for several confounding variables, it would have been beneficial to also adjust for other lifestyle factors such as diet and alcohol consumption, which can significantly influence health outcomes. In addition, information regarding medications or underlying conditions would have been beneficial to understand their potential impact on the observed results.

Follow-Up Methodology: The study does not seem to account for potential changes in participant lifestyle, drug use, anthropometric variables, or underlying conditions over the follow-up period, which could affect results. Anthropometric measurements and health status can change significantly over time, particularly in an ageing population. This should be discussed as a limitation and recommendation for future research that aim to incorporate repeated measures over time to capture this dynamic aspect of health should be stated.

Data Interpretation and Consistency: There appears to be a discrepancy in the interpretation of the results, specifically regarding the Kaplan-Meier survival curves and the conclusion. These sections suggest conflicting information about the effect of a lower thigh-hip ratio on mortality, which can be confusing to readers. This inconsistency needs to be resolved, or at least clarified, in order to support the validity of the study's findings.

Thigh-Hip Ratio vs. Thigh Circumference: The authors have chosen to focus on the thigh-hip ratio rather than thigh circumference alone, which is a unique and interesting approach. This allows for the consideration of body composition and proportionality, rather than focusing on a singular dimension of body size. However, it would be helpful for the authors to justify this choice more explicitly, particularly given their findings about the associations with normal BMI. They do not delve into a detailed discussion about why a high thigh-hip ratio would be beneficial for people with a normal BMI. While they discuss the role of thigh size as a measure of muscle and subcutaneous fat, the relevance of the hip measurement in this ratio is not extensively discussed. It seems from their findings that given BMI lower than 25kg/m2, a lower thigh circumference mainly reflects low muscle mass and emaciation. This is both logical and physiological. On that line, the addition of hip circumference seems to add more confusion than clarity. The authors should try to justify why it was necessary to include hip circumference.

Conclusion and Underlying Mechanisms: The paper concludes that a reduced thigh-hip ratio and a lower BMI are associated with increased all-cause mortality, but the factors contributing to this relationship remain unclear. This conclusion indicates that further research is required to understand the underlying biological mechanisms behind these associations. These should be added on the discussion, recommendation, or discussion section to improve clarity.

Reviewers' comments:

Reviewer's Responses to Questions

**Comments to the Author**

1. Is the manuscript technically sound, and do the data support the conclusions?

Reviewer #1: Yes

Reviewer #2: Yes

2. Has the statistical analysis been performed appropriately and rigorously? 

Reviewer #1: Yes

Reviewer #2: Yes

3. Have the authors made all data underlying the findings in their manuscript fully available?

Reviewer #1: No

Reviewer #2: No

4. Is the manuscript presented in an intelligible fashion and written in standard English?

Reviewer #1: Yes

Reviewer #2: Yes

5. Review Comments to the Author

Reviewer #1: General comments

Overall the article is well-written and will advance the body of scientific knowledge. However, the current form needs to be revised before the article can be published.

The followings need to be addressed in each section of the manuscript

1. Abstract

It is well-written and concisely summarizes the study's main findings.

2. Introduction

Well-presented and organized

3. Methodology

A revision is required for the methods section. The following section needs to be changed by the authors.

(a) Evaluation of risk factors

- “Thigh-to-waist (thigh–waist) ratio was determined by dividing waist circumference by thigh circumference” Revise this sentence for clearness and consistency

- Authors should mention the Names, Brands and City of manufactures for all

Types of equipment and tools that were used in measuring all variables.

- Similarly, the posture and any other instructions that were provided to the participants before taking weight and height should be mentioned. (for instance, were the participants instructed to remove clothes and shoes before taking weight and height measurements)

- “The quantification of alcohol intake was conducted using Japanese alcoholic beverage units” Insert the reference for this statement.

4. Results.

- why did authors categorize exercise as Yes/No? what tool did they use to collect information? about exercise.

- Table 3 baseline characteristics, what is AUR?

-Figure 2. What does the sentence below (1-specificity on X- axis) mean? And why should two languages be mixed?

5. Discussions

- I advise expanding the discussion to include other variables that were measured, analyzed but not discussed like, eGFR, HDL and LDL cholesterol and SUA.

-The last paragraph- revise sentences about confounding variables as limitations and strengths of the study concurrently.

Reviewer #2: Reviewer Comments

ABSTRACT

Well summarized and easily readable and understandable

INTRODUCTION

Well narrowed to the aim of the study

MATERIAL AND METHODS

Study design and participants

Physical activity: which type/kind of questionnaire was used? (Adopted or formulated).Exercise classification of yes and no is not clear, especially in quantifying the level of physical exercise. I recommend the standard classification by WHO or the American heart association guide may be used

On the measurement of anthropometrics: which instruments were used, the manufacture, brand, and model. How were the participants positioned when taking the measurements, was the type of clothes, shoes, and other accompanying object were they put into account on weight measurements (were they taken off or wearing light clothes)? All in all, what was the protocol/procedure for taking the anthropometrics?

Evaluation of risk factors

The statement: thigh to waist (thigh-waist) ratio was determined by dividing waist circumference by thigh circumference- not clear (may be reversed in one way)

On blood pressure measurement how was the participant positioned (sitting or supine) and what manufacture, brand, and model of the blood pressure machine used

How were the samples collected, stored, and measured and what type of machine/analyzer was used to assess these risk factors: Triglycerides (TG), High-density lipoprotein cholesterol (HDL-C), Low-density lipoprotein cholesterol (LDL-C), Serum uric acid (SUA), Hemoglobin A1c (HbA1c) and serum creatinine.

How were the cardiovascular diseases (ischemic heart disease, ischemic stroke, and peripheral vascular disease) determined was it by clinical or by investigation (for example ECG,

Note: indicate the source of the procedure for measurement and cut-off values for diagnosis of different risk factors

STATISTICAL ANALYSIS

BMI is divided into <25kg/m2 and >25 kg/m2, why not in the normal standard categorization of BMI such as <18.5 underweight, 18.5-24.9 normal, 25-29.9 overweight, 30-39.9 obese, and ≥40 very obese.

Physical exercise is classified as yes or no why, this does not give the ability to quantify the level of physical activity especially as recommended by the World Health Organization (WHO)

RESULTS

This statement on page 13 ‘’Our current findings align with our previous results, reaffirming the association between an elevated thigh-hip ratio and a heightened risk of all-cause mortality, particularly among individuals with a BMI < 25 kg/m2 and a sedentary lifestyle’’ not clear, which results are being referred? can the author please state or elaborate

These statements seem to contradict in trying to explain the impact of smaller/lower thigh circumference, hip circumference, thigh–hip ratio, and thigh-waist ratio in relation to the deceased and the cause of all mortality. i.e., statement 1: smaller thigh circumference and thigh hip ratio associated with deceased. Statement 2: lowest thigh-hip ratio showed a reduced risk of mortality and statement 3: elevated thigh-hip ratio and heightened risk associated with all causes of mortality: can the author please clarify these contradictions?

1. Page 11‘’the deceased participants had a significantly smaller thigh circumference and thigh–hip ratio (Table 2). This finding was true for both gender groups. Males who died had a significantly smaller thigh–waist ratio and females had a significantly smaller hip circumference

2. Page 12 ‘’The findings revealed consistent association patterns across both genders, demonstrating that individuals in the lowest thigh-hip ratio category exhibited a significantly reduced risk of mortality compared to those in all other categories’’

3. Page 13 ‘’Our current findings align with our previous results, reaffirming the association between an elevated thigh-hip ratio and a heightened risk of all-cause mortality, particularly among individuals with a BMI < 25 kg/m2 and a sedentary lifestyle’’

DISCUSSION

This statement, page 14 ‘’This study showed that men with a thigh circumference and thigh-hip ratio smaller than their hips have a higher risk of all-cause mortality, especially if their BMI is less than 25 kg/m2’’is not clear what the author met was the thigh circumference and thigh hip ratio smaller than their hips what? Circumference? what was the intended parameter that was being compared?

Since there was no standard categorization of BMI, Hence the quantification and linkage of obesity and overweight to the all-cause of mortality become unrealistic. Therefore, I recommend the author categorize based on the standard for easy quantification and linkage to the all-cause of mortality as a contributing factor.

The two statements below contradict each other, as the study was supposed to take into account the potential confounders in order to give the thigh-hip ratio the power to associate in all-cause mortality. Due to this most deaths may probably be due to those confounders found to be significant at baseline

‘’The inclusion of adjustments for various confounding variables and a sensitivity analysis, along with the measurement of anthropometric indices, are further advantages’’

‘’the study did not take into account confounding variables, drugs, underlying conditions, and lifestyle changes both at the baseline and throughout the follow-up period’’

Due to the presence of potential confounders such as age, systolic Blood Pressure, hypertension, chronic kidney disease, estimated Glomerular filtration rate, and proteinuria to cause mortality and were significant on the baseline. I recommend the author to expand the discussion.

CONCLUSION

May need some modification

6. PLOS authors have the option to publish the peer review history of their article (what does this mean?). If published, this will include your full peer review and any attached files.

Reviewer #1: No

Reviewer #2: **Yes: **Tuntufyege Erasto Mwasanjobe

---

## [Author Response · Author response to Decision Letter 0]

30 Jul 2023

PONE-D-23-14105

PLOS ONE

Dear Sir;　　

 Thank you very much for the valuable suggestions and comments on our manuscript entitled “Thigh-hip ratio is significantly associated with all-cause mortality among Japanese community-dwelling men”. We appreciate your positive suggestions on our manuscript. We are submitting here the revised manuscript. We revised it according to all the suggestions made by the reviewers, and the revisions are highlighted in red in the manuscript. Our incorporation of the reviewers’ suggestion is as follows:

Response 

Thank you for your advice and we met our manuscript PLOS ONE's style requirements.

Response

Thank you for your advice and we provided the correct grant numbers for the awards our received for our study in the ‘Funding Information’ section.

“We express our gratitude to Uni-edit (https://uni-edit.net/) for their valuable assistance in editing and proofreading this manuscript. This research received partial financial support from a grant-in-aid provided by the Foundation for Development of Community (2023). It is important to note that the funders played no part in the study design, data collection and analysis, decision to publish, or manuscript preparation.”

“No. This work was supported in part by a grant-in-aid from the Foundation for Development of Community (2023). No additional external funding was received for this study. The funders played no role in the study design, data collection and analysis, decision to publish, or manuscript preparation.”

Response

Thank you for your advice and the following text was removed from Acknowledgments. (Line 401-403)

Acknowledgments

We thank Uni-edit (https://uni-edit.net/) for editing and proofreading this manuscript.

Response

Thank you for your advice and we upload the minimal anonymized data set necessary to replicate our study findings as supporting information.

Response

Thank you for your advice and we corrected it.

Response

Yes. Our ethics statement included in the Methods. (Line 85-87)

The study protocol was examined and approved by the Ehime University Hospital’s Institutional Review Board (IRB: 1402009). All participants gave their written informed consent.

7. Please include your tables as part of your main manuscript and remove the individual files. Please note that supplementary tables (should remain/ be uploaded) as separate "supporting information" files

Response: Thank you for your advice and we included our tables as part of our main manuscript and remove the individual files.

Additional Editor Comments:

8. In this study the authors aimed to investigate the contributions of anthropometric measurements, independent of body mass index, to measures of all-cause mortality. They examined data for 1,704 participants from the 2014 Nomura Cohort Study who attended follow-ups for the subsequent eight years (follow-up rate: 93.0%).

This paper presents an intriguing study into the potential association of thigh-hip ratio with all-cause mortality in a Japanese population. There are, however, several areas that can benefit from more precise explanation, and there are certain limitations in the research that need to be addressed.

Study Design and Execution: The research is commendable for its prospective design, the extensive duration of the follow-up period, and the inclusion of sensitivity analyses and adjustment for confounding variables. The use of different anthropometric indices and their inclusion in the analysis provides a multi-dimensional perspective. The Kaplan-Meier survival curves were also useful tools in understanding the relationship between thigh-hip ratio and all-cause mortality. While the authors adjusted for several confounding variables, it would have been beneficial to also adjust for other lifestyle factors such as diet and alcohol consumption, which can significantly influence health outcomes. In addition, information regarding medications or underlying conditions would have been beneficial to understand their potential impact on the observed results.

Response

Thank you for your comments.

9. Follow-Up Methodology: The study does not seem to account for potential changes in participant lifestyle, drug use, anthropometric variables, or underlying conditions over the follow-up period, which could affect results. Anthropometric measurements and health status can change significantly over time, particularly in an ageing population. This should be discussed as a limitation and recommendation for future research that aim to incorporate repeated measures over time to capture this dynamic aspect of health should be stated.

Response:

Thank you for your advice and we described them as one of limitations in our study. (Line 376-383)

Third, we did not account for confounding variables, drugs, underlying conditions, and lifestyle changes at both baseline and throughout the follow-up period, which could affect results. Anthropometric measurements and health status can change significantly over time, particularly in an ageing population. Fourth, there was a lack of direct data concerning muscle mass, physical activity, and body composition. Collecting firsthand and repeated information regarding these aspects could offer valuable insights for a more comprehensive understanding of the connection between anthropometric parameters and mortality.

10. Data Interpretation and Consistency: There appears to be a discrepancy in the interpretation of the results, specifically regarding the Kaplan-Meier survival curves and the conclusion. These sections suggest conflicting information about the effect of a lower thigh-hip ratio on mortality, which can be confusing to readers. This inconsistency needs to be resolved, or at least clarified, in order to support the validity of the study's findings.

Response

Thank you for your advice and we corrected the presentation. (Line 239; 244-246; 270-273)

A smaller thigh–hip ratio was associated with a greater risk of all-cause mortality.

11. Thigh-Hip Ratio vs. Thigh Circumference: The authors have chosen to focus on the thigh-hip ratio rather than thigh circumference alone, which is a unique and interesting approach. This allows for the consideration of body composition and proportionality, rather than focusing on a singular dimension of body size. However, it would be helpful for the authors to justify this choice more explicitly, particularly given their findings about the associations with normal BMI. They do not delve into a detailed discussion about why a high thigh-hip ratio would be beneficial for people with a normal BMI. While they discuss the role of thigh size as a measure of muscle and subcutaneous fat, the relevance of the hip measurement in this ratio is not extensively discussed. It seems from their findings that given BMI lower than 25kg/m2, a lower thigh circumference mainly reflects low muscle mass and emaciation. This is both logical and physiological. On that line, the addition of hip circumference seems to add more confusion than clarity. The authors should try to justify why it was necessary to include hip circumference. 

Response

Thank you for your advice and we added the explanation in Discussion. (Line 350-366) 

Previous studies have indicated that gluteal subcutaneous fat and muscle mass are assessed through hip circumference [38], while waist circumference primarily indicates subcutaneous and visceral adipose tissue levels. Interestingly, hip circumference shows a strong correlation with BMI, and there is also a correlation between BMI and waist circumference [39]. Hence, a larger hip circumference could suggest higher BMI and waist circumference. To obtain a more accurate estimation of the positive effects of femoral muscle mass, it might be beneficial to adjust thigh circumference (which reflects beneficial fat and muscle mass) in relation to hip circumference or waist circumference (which includes harmful visceral fat). Furthermore, this approach may more accurately assess the benefit of thigh muscle mass for male participants with lower BMIs. These sex-specific differences imply that the biological processes causing these disparities may be influenced by genes, sex hormones, and gender-based differences in hip size, body composition, and/or fat distribution [37]. However, the factors contributing to this relationship remain unknown. Thus, further research is required to understand the underlying biological mechanisms behind these associations.

12. Conclusion and Underlying Mechanisms: The paper concludes that a reduced thigh-hip ratio and a lower BMI are associated with increased all-cause mortality, but the factors contributing to this relationship remain unclear. This conclusion indicates that further research is required to understand the underlying biological mechanisms behind these associations. These should be added on the discussion, recommendation, or discussion section to improve clarity.

Response

Thank you for your advice and we added the explanation in the discussion to improve clarity. (Line 340-366)

Previous studies have indicated that smaller-than-expected hip circumferences are related to reduced femoral fat, tiny pelvic bone structure, or muscle atrophy in the gluteofemoral region [33]. Conversely, larger hip and thigh circumferences are associated with more favorable lipid and glucose levels [8] [34] [35]. When not accounting for hip circumference, the impact of central obesity on mortality risk is grossly underestimated. A Framingham-type model for CVD mortality showed a significant boost in prediction power when waist and hip circumference were included [36]. While, the People’s Republic of China Study on 1,144 male and 1,776 female participants reported that a larger hip circumference was not associated with reduced incident risk factors in male participants [37]. Our study also did not link hip circumference with all-cause mortality. Previous studies have indicated that gluteal subcutaneous fat and muscle mass are assessed through hip circumference [38], while waist circumference primarily indicates subcutaneous and visceral adipose tissue levels. Interestingly, hip circumference shows a strong correlation with BMI, and there is also a correlation between BMI and waist circumference [39]. Hence, a larger hip circumference could suggest higher BMI and waist circumference. To obtain a more accurate estimation of the positive effects of femoral muscle mass, it might be beneficial to adjust thigh circumference (which reflects beneficial fat and muscle mass) in relation to hip circumference or waist circumference (which includes harmful visceral fat). Furthermore, this approach may more accurately assess the benefit of thigh muscle mass for male participants with lower BMIs. These sex-specific differences imply that the biological processes causing these disparities may be influenced by genes, sex hormones, and gender-based differences in hip size, body composition, and/or fat distribution [37]. However, the factors contributing to this relationship remain unknown. Thus, further research is required to understand the underlying biological mechanisms behind these associations.

Reviewers' comments:

Reviewer's Responses to Questions

Comments to the Author

1. Is the manuscript technically sound, and do the data support the conclusions?

Reviewer #1: Yes

Reviewer #2: Yes

2. Has the statistical analysis been performed appropriately and rigorously?

Reviewer #1: Yes

Reviewer #2: Yes

3. Have the authors made all data underlying the findings in their manuscript fully available?

Reviewer #1: No

Reviewer #2: No

4. Is the manuscript presented in an intelligible fashion and written in standard English?

Reviewer #1: Yes

Reviewer #2: Yes

5. Review Comments to the Author

Reviewer #1: General comments

Overall the article is well-written and will advance the body of scientific knowledge. However, the current form needs to be revised before the article can be published.

The followings need to be addressed in each section of the manuscript

1. Abstract

It is well-written and concisely summarizes the study's main findings.

Response

Thank you for your comments.

2. Introduction

Well-presented and organized

Response

Thank you for your comments.

3. Methodology

A revision is required for the methods section. The following section needs to be changed by the authors.

(a) Evaluation of risk factors

- “Thigh-to-waist (thigh–waist) ratio was determined by dividing waist circumference by thigh circumference” Revise this sentence for clearness and consistency

Response

Thank you for your advice and we corrected this sentence for clearness and consistency. (Line 97-109)

All anthropometric measurements were in accordance with the WHO standards [21]. Body weight was measured to the nearest 0.1 kg with an electronic standard weight scale machine, with participants wearing underwear or light clothing and body height was measured without shoes to the nearest 1 cm using a stadiometer. Body mass index (BMI) was determined as weight (kg) divided by squared height (m2). Hip circumference was measured at the widest level over the greater trochanters, and waist circumference was estimated in the horizontal plane at the midpoint between the anterior iliac crest and the lower edge of the ribs. Thigh circumference was measured just below the gluteal fold on both legs and the mean of the two was used in the analysis. Thigh-to-waist (thigh–waist) ratio was computed by dividing thigh circumference by waist circumference, and waist-to-hip (waist–hip) ratio by dividing waist circumference by hip circumference.

4. Authors should mention the Names, Brands and City of manufactures for all

Types of equipment and tools that were used in measuring all variables.

Response

Thank you for your advice and we mentioned the Names, Brands and City of manufactures for all types of equipment and tools. (Line 118-128)

Systolic and diastolic blood pressure (SBP and DBP, respectively) were measured using an automated sphygmomanometer (BP-103i; Colin, Aichi, Japan) with an appropriately sized cuff placed on their right upper arm after at least 5 minutes of immobility, and the average of two consecutive measurements was used for calculation. Participants were asked to fast overnight before measurements and their triglycerides (TG), high-density lipoprotein cholesterol (HDL-C), low-density lipoprotein cholesterol (LDL-C), serum uric acid (SUA), and creatinine (Cr) (enzymatic assay, Kyowa Medex, Tokyo, Japan) were measured using an automated analyzer (Hitachi, Tokyo, Japan). Hemoglobin A1c (HbA1c) (latex agglutination inhibition assay, Kyowa Medex, Tokyo, Japan) was measured using an automated analyzer (DM-JACK® Ex, Tokyo, Japan). 

5. Similarly, the posture and any other instructions that were provided to the participants before taking weight and height should be mentioned. (for instance, were the participants instructed to remove clothes and shoes before taking weight and height measurements)

Response

Thank you for your advice and we inserted this sentence. (Line 97-101)

All anthropometric measurements were in accordance with the WHO standards [21]. Body weight was measured to the nearest 0.1 kg with an electronic standard weight scale machine, with participants wearing underwear or light clothing and body height was measured without shoes to the nearest 1 cm using a stadiometer. 

6. “The quantification of alcohol intake was conducted using Japanese alcoholic beverage units” Insert the reference for this statement.

Response

Thank you for your advice and we inserted the reference. (Line 114-118)

The amount of alcohol consumed each day was calculated using Japanese alcoholic beverage units (22.9 g ethanol), and participants were divided into four groups: nondrinkers, occasional drinkers (< 1 unit/day), moderate daily drinkers (1–2 units/day), and heavy daily drinkers (3 units/day); none drank more than 3 units/day [22].

4. Results.

- why did authors categorize exercise as Yes/No? what tool did they use to collect information? about exercise.

Response:

Thank you for your advice and we inserted the explanation of

exercise in detail. (Line 94- 97)

To qualify as having exercise habits, the participants must have participated in any type of moderate-to-vigorous physical activity, such as brisk walking, golfing, gardening, jogging, or playing tennis, for at least 30 minutes on at least 2 days per week for a minimum of 1 year.

5. Table 3 baseline characteristics, what is AUR?

Response 

Thank you for your advice and we inserted an area under the curve (AUR) in text. (Line 215-219)

Table 3 and Fig 2 present the gender-specific area under the ROC curve (AUC) for individual anthropometric indices used to predict all-cause mortality. Thigh–hip ratio yielded the lowest AUC for males (0.311, 95% confidence interval [CI]: 0.251–0.371), and thigh circumference yielded the lowest AUC for females (0.356, 95% CI: 0.283–0.430).

6. Figure 2. What does the sentence below (1-specificity on X- axis) mean? And why should two languages be mixed?

Response

Thank you for your advice and we corrected it. (Figure2))

6. Discussions

- I advise expanding the discussion to include other variables that were measured, analyzed but not discussed like, eGFR, HDL and LDL cholesterol and SUA.

Response

Thank you for your advice and we discussed them in Discussion section. (Line 286-297; Line 393-399)

A key conclusion of this cohort study is that the anthropometric measure of thigh–hip ratio is a substantial and independent predictor of all-cause mortality in community-dwelling men, even after adjustment for age, BMI, smoking status, drinking status, exercise status, history of CVD, hypertension, low HDL-cholesterolemia, high LDL-cholesterolemia, hypertriglyceridemia, diabetes, CKD, hyperuricemia at baseline. Smaller thigh–hip ratio was associated with death, however, only in individuals with BMI < 25 kg/m2. The study also highlighted an interaction between thigh–hip ratio and BMI in the context of all-cause mortality. To the best of our knowledge, this study is the first to demonstrate an association between thigh–hip ratio and all-cause mortality in community-dwelling men in Japan. However, it is possible that some important confounding factors were not identified or adjusted for.

They demonstrate that, after correcting for potential confounding variables such as age, body mass index, smoking status, drinking status, exercise status, history of cardiovascular disease, hypertension, low HDL-cholesterolemia, high LDL-cholesterolemia, hypertriglyceridemia, diabetes, CKD, and hyperuricemia at baseline, a smaller thigh–hip ratio in males and a lower BMI in females predict increased all-cause mortality.

7.The last paragraph- revise sentences about confounding variables as limitations and strengths of the study concurrently.

Response

Thank you for your advice and we corrected it. (Line 376-383)

Third, we did not account for confounding variables, drugs, underlying conditions, and lifestyle changes at both baseline and throughout the follow-up period, which could affect results. Anthropometric measurements and health status can change significantly over time, particularly in an ageing population. Fourth, there was a lack of direct data concerning muscle mass, physical activity, and body composition. Collecting firsthand and repeated information regarding these aspects could offer valuable insights for a more comprehensive understanding of the connection between anthropometric parameters and mortality.

Reviewer #2: Reviewer Comments

ABSTRACT

1. Well summarized and easily readable and understandable

 Response

Thank you for your comments.

2. INTRODUCTION

Well narrowed to the aim of the study

Response

Thank you for your comments.

MATERIAL AND METHODS

3. Study design and participants

Physical activity: which type/kind of questionnaire was used? (Adopted or formulated).Exercise classification of yes and no is not clear, especially in quantifying the level of physical exercise. I recommend the standard classification by WHO or the American heart association guide may be used.

Response: Thank you for your advice and we have inserted about exercise habits

in Evaluation of risk factors of Methods. (Line 92-97)

The participants' current status, their level of physical activity (e.g., exercise routines), details about their medical history, and medications were collected through a structured questionnaire interview. To qualify as having exercise habits, the participants must have participated in any type of moderate-to-vigorous physical activity, such as brisk walking, golfing, gardening, jogging, or playing tennis, for at least 30 minutes on at least 2 days per week for a minimum of 1 year.

4. On the measurement of anthropometrics: which instruments were used, the manufacture, brand, and model. How were the participants positioned when taking the measurements, was the type of clothes, shoes, and other accompanying object were they put into account on weight measurements (were they taken off or wearing light clothes)? 

Response

Thank you for your advice and we have inserted what to wear for weigh-ins in

Evaluation of risk factors of Methods. (99-101)

Body weight was measured to the nearest 0.1 kg with an electronic standard weight scale machine, with participants wearing underwear or light clothing and Body height was measured without shoes to the nearest 1 cm using a stadiometer.

5. All in all, what was the protocol/procedure for taking the anthropometrics?

 Response

Thank you for your advice and we added the protocol/procedure for taking the anthropometrics? (Line 97-110)

All anthropometric measurements were in accordance with the WHO standards [21]. Body weight was measured to the nearest 0.1 kg with an electronic standard weight scale machine, with participants wearing underwear or light clothing and body height was measured without shoes to the nearest 1 cm using a stadiometer. Body mass index (BMI) was determined as weight (kg) divided by squared height (m2). Hip circumference was measured at the widest level over the greater trochanters, and waist circumference was estimated in the horizontal plane at the midpoint between the anterior iliac crest and the lower edge of the ribs. Thigh circumference was measured just below the gluteal fold on both legs and the mean of the two was used in the analysis. Thigh-to-waist (thigh–waist) ratio was computed by dividing thigh circumference by waist circumference, and waist-to-hip (waist–hip) ratio by dividing waist circumference by hip circumference. Similarly, the thigh to hip (thigh–hip) ratio was calculated by dividing thigh circumference by hip circumference.

6. The statement: thigh to waist (thigh-waist) ratio was determined by dividing waist circumference by thigh circumference- not clear (may be reversed in one way)

 Response

Thank you for your advice and we corrected it. (Line 107-108)

Thigh-to-waist (thigh–waist) ratio was computed by dividing thigh circumference by waist circumference.

7. On blood pressure measurement how was the participant positioned (sitting or supine) and what manufacture, brand, and model of the blood pressure machine used

 Response

Thank you for your advice and we have inserted blood pressure measurement

method in Evaluation of risk factors of Methods. (Line 118-122)

Systolic and diastolic blood pressure (SBP and DBP, respectively) were measured using an automated sphygmomanometer (BP-103i; Colin, Aichi, Japan) after the participants had remained motionless for at least five minutes with a properly sized cuff placed on their right upper arm, and the average of two consecutive readings was used for calculation.

8. How were the samples collected, stored, and measured and what type of machine/analyzer was used to assess these risk factors: Triglycerides (TG), High-density lipoprotein cholesterol (HDL-C), Low-density lipoprotein cholesterol (LDL-C), Serum uric acid (SUA), Hemoglobin A1c (HbA1c) and serum creatinine. 

Response

Thank you for your advice and we added them in Methods

Participants were asked to fast overnight before measurements and their triglycerides (TG), high-density lipoprotein cholesterol (HDL-C), low-density lipoprotein cholesterol (LDL-C), serum uric acid (SUA), and creatinine (Cr) (enzymatic assay, Kyowa Medex, Tokyo, Japan) were measured using an automated analyzer (Hitachi, Tokyo, Japan). Hemoglobin A1c (HbA1c) (latex agglutination inhibition assay, Kyowa Medex, Tokyo, Japan) was measured using an automated analyzer (DM-JACK® Ex, Tokyo, Japan).

9. How were the cardiovascular diseases (ischemic heart disease, ischemic stroke, and peripheral vascular disease) determined was it by clinical or by investigation (for example ECG, Note: indicate the source of the procedure for measurement and cut-off values for diagnosis of different risk factors

Response

Thank you for your advice and we added the explanation. (Line 145-146)

Cardiovascular disease includes self-reported ischemic heart disease, stroke, and peripheral vascular disease.

STATISTICAL ANALYSIS

10. BMI is divided into <25kg/m2 and >25 kg/m2, why not in the normal standard categorization of BMI such as <18.5 underweight, 18.5-24.9 normal, 25-29.9 overweight, 30-39.9 obese, and ≥40 very obese.

Response

Thank you for your advice but the number of cases was divided by 25 kg/m2, because the number of cases of underweight and obese would be extremely small if BMI were divided by standardized BMI. (<18.5 kg/m2 underweight, 31 cases; 18.5-24.9 normal, 548 cases; 25-29.9 overweight, 167 cases; 30-39.9 obese, 19 cases). We added them to the limitations of the study. (Line 171; 384-386)

Fifth, in Japan, where the prevalence and severity of obesity is still mild, we examined the interaction between BMI and thigh–hip ratio on mortality in two groups based on a BMI of 25 or greater. However, the results are not necessarily generalizable.

11. Physical exercise is classified as yes or no why, this does not give the ability to quantify the level of physical activity especially as recommended by the World Health Organization (WHO)

Response

Thank you for your advice and we have inserted about exercise habits in

Evaluation of risk factors of Methods. (Line 92-97)

The participants' current status, their level of physical activity (e.g., exercise routines), details about their medical history, and medications were collected through a structured questionnaire interview. To qualify as having exercise habits, the participants must have participated in any type of moderate-to-vigorous physical activity, such as brisk walking, golfing, gardening, jogging, or playing tennis, for at least 30 minutes on at least 2 days per week for a minimum of 1 year.

RESULTS

12. This statement on page 13 ‘’Our current findings align with our previous results, reaffirming the association between an elevated thigh-hip ratio and a heightened risk of all-cause mortality, particularly among individuals with a BMI < 25 kg/m2 and a sedentary lifestyle’’ not clear, which results are being referred? can the author please state or elaborate

These statements seem to contradict in trying to explain the impact of smaller/lower thigh circumference, hip circumference, thigh–hip ratio, and thigh-waist ratio in relation to the deceased and the cause of all mortality. i.e., statement 1: smaller thigh circumference and thigh hip ratio associated with deceased. Statement 2: lowest thigh-hip ratio showed a reduced risk of mortality and statement 3: elevated thigh-hip ratio and heightened risk associated with all causes of mortality: can the author please clarify these contradictions?

Response

Thank you for your advice and we have corrected this sentence. (Line 270-272)

Similar to our previous results, a smaller thigh–hip ratio was associated with a greater risk of all-cause mortality, and this relationship was particularly pronounced among participants with BMI < 25 kg/m2.

1) Page 11‘’the deceased participants had a significantly smaller thigh circumference and thigh–hip ratio (Table 2). This finding was true for both gender groups. Males who died had a significantly smaller thigh–waist ratio and females had a significantly smaller hip circumference

2) Page 12 ‘’The findings revealed consistent association patterns across both genders, demonstrating that individuals in the lowest thigh-hip ratio category

3) Page 13 ‘’Our current findings align with our previous results, reaffirming the association between an elevated thigh-hip ratio and a heightened risk of all-cause mortality, particularly among individuals with a BMI < 25 kg/m2 and a sedentary lifestyle’’

exhibited a significantly reduced risk of mortality compared to those in all other categories’’

Response

Thank you for your advice and we have corrected this sentence. (Line 244-246; 273-240; 270-274)

The results indicated that the association patterns were similar for the thigh–hip ratio for both genders, and mortality risk was significantly greater for individuals in the smallest thigh–hip ratio category than for those in all other categories (p < 0.001 for both genders).

Fig. 3 Kaplan–Meier survival curves for four categories of thigh–hip ratio by gender. Thigh–hip ratio showed similar association patterns for both gender groups, and mortality risk was significantly greater for individuals with the smallest category of thigh–hip ratio than for those in all other categories (p < 0.001 for both genders).

Similar to our previous results, a smaller thigh–hip ratio was associated with a greater risk of all-cause mortality, and this relationship was particularly pronounced among participants with BMI < 25 kg/m2 and who did not exercise.

DISCUSSION

13. This statement, page 14 ‘’This study showed that men with a thigh circumference and thigh-hip ratio smaller than their hips have a higher risk of all-cause mortality, especially if their BMI is less than 25 kg/m2’’is not clear what the author met was the thigh circumference and thigh hip ratio smaller than their hips what? Circumference? what was the intended parameter that was being compared?

Response

Thank you for your advice and we corrected this sentence. (Line 312-316)

The present study showed that smaller thigh circumference and thigh–hip ratio are significantly associated with an increased risk of all-cause mortality in men. In other words, male health and survival were negatively impacted when their thigh-hip ratio was smaller, especially if they had a relatively low body weight (BMI < 25 kg/m2).

14. Since there was no standard categorization of BMI, Hence the quantification and linkage of obesity and overweight to the all-cause of mortality become unrealistic. Therefore, I recommend the author categorize based on the standard for easy quantification and linkage to the all-cause of mortality as a contributing factor.

Response

Thank you for your advice but the number of cases was divided by 25 kg/m2, because the number of cases of underweight and obese would be extremely small if BMI were divided by standardized BMI. (<18.5 kg/m2 underweight, 31 cases; 18.5-24.9 normal, 548 cases; 25-29.9 overweight, 167 cases; 30-39.9 obese, 19 cases). In addition, obesity is adequately specified as a BMI ≥ 25 kg/m2 in Japan where the prevalence and degree of obesity remains mild. We added them to the limitations of the study. (Line 383-387)

Fourth, there was a lack of direct data concerning muscle mass, physical activity, and body composition. Fifth, in Japan, where the prevalence and severity of obesity is still mild, we examined the interaction between BMI and thigh–hip ratio on mortality in two groups based on a BMI of ≥ 25 kg/m2. However, the results are not necessarily generalizable. 

15. The two statements below contradict each other, as the study was supposed to consider the potential confounders in order to give the thigh-hip ratio the power to associate in all-cause mortality. Due to this most deaths may probably be due to those confounders found to be significant at baseline. ‘’The inclusion of adjustments for various confounding variables and a sensitivity analysis, along with the measurement of anthropometric indices, are further advantages’’

Response

Thank you for your comments.

16. ‘’the study did not take into account confounding variables, drugs, underlying conditions, and lifestyle changes both at the baseline and throughout the follow-up period’’ Due to the presence of potential confounders such as age, systolic Blood Pressure, hypertension, chronic kidney disease, estimated Glomerular filtration rate, and proteinuria to cause mortality and were significant on the baseline. I recommend the author to expand the discussion.

Response

Thank you for your advice and we corrected it. (Line 286-297; 276-283)

A key conclusion of this cohort study is that the anthropometric measure of thigh–hip ratio is a substantial and independent predictor of all-cause mortality in community-dwelling men, even after adjustment for age, BMI, smoking status, drinking status, exercise status, history of CVD, hypertension, low HDL-cholesterolemia, high LDL-cholesterolemia, hypertriglyceridemia, diabetes, CKD, hyperuricemia at baseline. Smaller thigh–hip ratio was associated with death, however, only in individuals with BMI < 25 kg/m2. The study also highlighted an interaction between thigh–hip ratio and BMI in the context of all-cause mortality. To the best of our knowledge, this study is the first to demonstrate an association between thigh–hip ratio and all-cause mortality in community-dwelling men in Japan. However, it is must be considered that some important confounding factors were not identified or adjusted for.

Third, we did not account for confounding variables, drugs, underlying conditions, and lifestyle changes at both baseline and throughout the follow-up period, which could affect results. Anthropometric measurements and health status can change significantly over time, particularly in an ageing population. Fourth, there was a lack of direct data concerning muscle mass, physical activity, and body composition. Collecting firsthand and repeated information regarding these aspects could offer valuable insights for a more comprehensive understanding of the connection between anthropometric parameters and mortality.

CONCLUSION

14. May need some modification

Response 

Thank you for your advice and we corrected it. (Line 393-399) 

The present conclusions are based on an eight-year follow-up study of adults (≥ 20 years). They demonstrate that, after correcting for potential confounding variables such as age, body mass index, smoking status, drinking status, exercise status, history of cardiovascular disease, hypertension, low HDL-cholesterolemia, high LDL-cholesterolemia, hypertriglyceridemia, diabetes, CKD, and hyperuricemia at baseline, a smaller thigh–hip ratio in males and a lower BMI in females predict increased all-cause mortality. 

In summary, we appreciate your careful reading of our manuscript and instructive comments. We believe that our revised manuscript has been improved by the incorporation your suggestions. We hope that the changes will meet with your approval.

---

## [Decision Letter · Decision Letter 1]

15 Aug 2023

PONE-D-23-14105R1Thigh-hip ratio is significantly associated with all-cause mortality among Japanese community-dwelling menPLOS ONE

Dear Dr. Kawamoto,

Thank you for submitting your manuscript to PLOS ONE. After careful consideration, we feel that it has merit but does not fully meet PLOS ONE’s publication criteria as it currently stands. Therefore, we invite you to submit a revised version of the manuscript that addresses the points raised during the review processPlease address thoroughly and completely  all the concerns from the reviewersThis manuscript requires a minor but THOROUGH reviewMost of the previously raised comments have been addressed, but there are still some minor but important clarifications and improvements to be done

We look forward to receiving your revised manuscript.

Kind regards,

Fredirick Lazaro mashili, MD, PhD

Academic Editor

PLOS ONE

Journal Requirements:

Additional Editor Comments:

Please clearly and thoroughly address all the comments raised by reviewers

Reviewers' comments:

Reviewer's Responses to Questions

**Comments to the Author**

1. If the authors have adequately addressed your comments raised in a previous round of review and you feel that this manuscript is now acceptable for publication, you may indicate that here to bypass the “Comments to the Author” section, enter your conflict of interest statement in the “Confidential to Editor” section, and submit your "Accept" recommendation.

Reviewer #1: (No Response)

Reviewer #2: (No Response)

2. Is the manuscript technically sound, and do the data support the conclusions?

Reviewer #1: Yes

Reviewer #2: Yes

3. Has the statistical analysis been performed appropriately and rigorously? 

Reviewer #1: Yes

Reviewer #2: Yes

4. Have the authors made all data underlying the findings in their manuscript fully available?

Reviewer #1: No

Reviewer #2: Yes

5. Is the manuscript presented in an intelligible fashion and written in standard English?

Reviewer #1: Yes

Reviewer #2: Yes

6. Review Comments to the Author

Reviewer #1: Additional comments to be addressed

1. Line 92 "The participants' current status, their level of physical activity (e.g., exercise habits)". Revise the sentence as exercise habits is not an example of levels of physical activity (levels of physical activity range from sedentary to vigorous)

2. It is important for the authors to note that exercise habits are not mutually exclusive. In other words, you cannot classify exercise habits as binary "Yes or NO". Levels of physical activity range from sedentary to vigorous. Please revise the classification of levels of physical activity in entire manuscript.

3. Line 122-128 the paragraph is not clear.

4. Line 141 "Participants were classified as having hypertriglyceridemia if their TG level was < 150 mg/dL" This is not correct.

Reviewer #2: Reviewer Comments for the revised manuscript

ABSTRACT

Well summarized

INTRODUCTION

Narrowed to the aim of the study

MATERIAL AND METHODS

Study design and participants

Line 92-97 is clear but does not quantify the level of physical activity to meet the minimum requirement.

The brand, model, and manufacturer of the instrument used are not stated (the weighing scale and the stadiometer) in lines 99 and 101. Also, the instrument for measuring waist, hip, and thigh circumference is not stated, in lines 103-106

Evaluation of risk factors

Line 107-108 well revised and is clear

Lines 118-122 are well revised, but the positioning of participants is not stated, can the author state the position

Line 145-146 is clear cardiovascular disease

STATISTICAL ANALYSIS

Lines 383-387 state the limitation of the study of categorizing BMI of 25 or greater

Lines 92-97: explanation is clear but does not quantify the level of physical activity to meet the minimum requirement, the recommended is 600MET/MIN/WEEK for healthy living

RESULTS

Lines 270-272, 244-246, 240-273,270: are clear well revised the comments put forth on the previous manuscript

DISCUSSION

Lines 312-316 are clear and understood

Lines 383-387 are clear are limitation factors for BMI categorization

Line 276-283,286-297: noted for elaboration on the confounders

CONCLUSION

Line 393-399: noted for the modification done.

Thank the response to the comments given

7. PLOS authors have the option to publish the peer review history of their article (what does this mean?). If published, this will include your full peer review and any attached files.

Reviewer #1: **Yes: **Oscar Mbembela

Reviewer #2: No

---

## [Author Response · Author response to Decision Letter 1]

19 Aug 2023

PONE-D-23-14105

PLOS ONE

Dear Sir;　　

 Thank you very much for the valuable suggestions and comments on our manuscript entitled “Thigh-hip ratio is significantly associated with all-cause mortality among Japanese community-dwelling men”. We appreciate your positive suggestions on our manuscript. W

e are submitting here the revised manuscript. We revised it according to all the suggestions made by the reviewers, and the revisions are highlighted in red in the manuscript. Our incorporation of the reviewers’ suggestion is as follows:

 Response

Thank you for your advice and we have confirmed them.

Journal Requirements:

 Response

Thank you for your advice and we have confirmed them.

Additional Editor Comments:

Please clearly and thoroughly address all the comments raised by reviewers

6. Review Comments to the Author

Reviewer #1: Additional comments to be addressed

1. Line 92 "The participants' current status, their level of physical activity (e.g., exercise habits)". Revise the sentence as exercise habits is not an example of levels of physical activity (levels of physical activity range from sedentary to vigorous)

 Response

Thank you for your advice and we corrected it. (Line 92-93)

The participants' current status, exercise habits, details about their medical history, and medications were collected through a structured questionnaire interview.

2. It is important for the authors to note that exercise habits are not mutually exclusive. In other words, you cannot classify exercise habits as binary "Yes or NO". Levels of physical activity range from sedentary to vigorous. Please revise the classification of levels of physical activity in entire manuscript.

Response

Thank you for your advice and in this study, we classified those who have been exercising continuously for a certain period of time as having an exercise habit, as shown in the examples. Thus, we only have data on the presence or absence of exercise habits. We mentioned its importance in the limitations of the study. (Line 92-98; Line 385-387)

 To qualify as having exercise habits, the participants must have participated in any type of moderate-to-vigorous physical activity, such as brisk walking, golfing, gardening, jogging, or playing tennis, for at least 30 minutes on at least 2 days per week (≥ 600 MET-minutes/week) for a minimum of 1 year.

Fourth, there was a lack of direct and detail data concerning exercise habits, muscle mass, and body composition which are important for maintaining physique and health.

3. Line 122-128 the paragraph is not clear.

 Response

Thank you for your advice and we corrected it. (Line 125-131)

Participants were asked to fast overnight before measurements and their triglycerides (TG), high-density lipoprotein cholesterol (HDL-C), low-density lipoprotein cholesterol (LDL-C), serum uric acid (SUA), and creatinine (Cr) (enzymatic assay, Kyowa Medex, Tokyo, Japan) were measured using an automated analyzer (Hitachi, Tokyo, Japan). Hemoglobin A1c (HbA1c) was analyzed by the latex agglutination inhibition method (Kyowa Medex, Tokyo, Japan) using an automated analyzer (DM-JACK® Ex, Tokyo, Japan).

4. Line 141 "Participants were classified as having hypertriglyceridemia if their TG level was < 150 mg/dL" This is not correct.

Response

Thank you for your advice and we corrected it. (Line 144)

.......and hypertriglyceridemia if their TG level was ≥ 150 mg/dL.

Reviewer #2: Reviewer Comments for the revised manuscript

1. ABSTRACT

Well summarized

Response：Thank you for your comment.

2. INTRODUCTION

Narrowed to the aim of the study

Response：Thank you for your comment.

3. MATERIAL AND METHODS

Study design and participants

3.1. Line 92-97 is clear but does not quantify the level of physical activity to meet the minimum requirement.

Response

Thank you for your advice but in this study, we classified those who have been exercising continuously for a certain period of time as having an exercise habit, as shown in the examples. Thus, we only have data on the presence or absence of exercise habits. We mentioned its importance in the limitations of the study. (Line 385-387)

Fourth, there was a lack of direct and detail data concerning exercise habits, muscle mass, and body composition which are important for maintaining physique and health.

3.2. The brand, model, and manufacturer of the instrument used are not stated (the weighing scale and the stadiometer) in lines 99 and 101. Also, the instrument for measuring waist, hip, and thigh circumference is not stated, in lines 103-106

Response

Thank you for your advice and we inserted them. (Line 99-105)

Body weight was measured to the nearest 0.1 kg with an electronic standard weight scale machine (HBF-214, Omron, Tokyo, Japan), with participants wearing underwear or light clothing and body height was measured without shoes to the nearest 1 cm using a stadiometer (DSN-90, Muratec-KDS, Kyoto, Japan). Body mass index (BMI) was determined as weight (kg) divided by squared height (m2). Waist, hip, and thigh circumferences (cm) were measured with an anthropometric tape (25-204, Clover, Osaka, Japan).

4. Evaluation of risk factors

4.1. Line 107-108 well revised and is clear

Response：Thank you for your comment.

4.2. Lines 118-122 are well revised, but the positioning of participants is not stated, can the author state the position

Response

Thank you for your advice and we corrected it. (Line 121-125)

Systolic and diastolic blood pressure (SBP and DBP, respectively) were measured using an automated sphygmomanometer (BP-103i; Colin, Aichi, Japan) with an appropriately sized cuff placed on their right upper arm while the subjects were seated after at least 5 minutes of immobility, and the average of two consecutive measurements was used for calculation.

4.3. Line 145-146 is clear cardiovascular disease

Response：Thank you for your comment.

5. STATISTICAL ANALYSIS

5.1. Lines 383-387 state the limitation of the study of categorizing BMI of 25 or greater

Response：Thank you for your comment.

5.2. Lines 92-97: explanation is clear but does not quantify the level of physical activity to meet the minimum requirement, the recommended is 600MET/MIN/WEEK for healthy living

Response

Thank you for your advice but in this study, we classified those who have been exercising continuously for a certain period of time as having an exercise habit, as shown in the examples. Thus, we only have data on the presence or absence of exercise habits. We mentioned its importance in the limitations of the study. (Line 94-98; Line 385-387)

To qualify as having exercise habits, the participants must have participated in any type of moderate-to-vigorous physical activity, such as brisk walking, golfing, gardening, jogging, or playing tennis, for at least 30 minutes on at least 2 days per week (≥ 600 MET-minutes/week) for a minimum of 1 year.

Fourth, there was a lack of direct and detail data concerning exercise habits, muscle mass, and body composition which are important for maintaining physique and health.

5. RESULTS

5.1. Lines 270-272, 244-246, 240-273,270: are clear well revised the comments put forth on the previous manuscript

Response：Thank you for your comment.

6. DISCUSSION

Lines 312-316 are clear and understood

Lines 383-387 are clear are limitation factors for BMI categorization

Line 276-283,286-297: noted for elaboration on the confounders

Response：Thank you for your comment.

7. CONCLUSION

Line 393-399: noted for the modification done.

Response：Thank you for your comment.

In summary, we appreciate your careful reading of our manuscript and instructive comments. We believe that our revised manuscript has been improved by the incorporation your suggestions. We hope that the changes will meet with your approval.

---

## [Decision Letter · Decision Letter 2]

18 Sep 2023

Thigh-hip ratio is significantly associated with all-cause mortality among Japanese community-dwelling men

PONE-D-23-14105R2

Dear Dr. Kawamoto,

We’re pleased to inform you that your manuscript has been judged scientifically suitable for publication and will be formally accepted for publication once it meets all outstanding technical requirements.

Kind regards,

Fredirick Lazaro mashili, MD, PhD

Academic Editor

PLOS ONE

Additional Editor Comments (optional):

All comments have been sufficiently addressed

Reviewers' comments:

Reviewer's Responses to Questions

**Comments to the Author**

1. If the authors have adequately addressed your comments raised in a previous round of review and you feel that this manuscript is now acceptable for publication, you may indicate that here to bypass the “Comments to the Author” section, enter your conflict of interest statement in the “Confidential to Editor” section, and submit your "Accept" recommendation.

Reviewer #1: All comments have been addressed

Reviewer #3: All comments have been addressed

2. Is the manuscript technically sound, and do the data support the conclusions?

Reviewer #1: Yes

Reviewer #3: Yes

3. Has the statistical analysis been performed appropriately and rigorously? 

Reviewer #1: Yes

Reviewer #3: Yes

4. Have the authors made all data underlying the findings in their manuscript fully available?

Reviewer #1: Yes

Reviewer #3: Yes

5. Is the manuscript presented in an intelligible fashion and written in standard English?

Reviewer #1: No

Reviewer #3: Yes

6. Review Comments to the Author

Reviewer #1: Authors have adequately addressed all comments raised during my previous round of review. Therefore I suggest accepting the manuscript for publication.

Reviewer #3: The authors have addressed all the queries raised by the reviewers, and when necessary, they have also acknowledged limitations in addressing certain comments fully.

7. PLOS authors have the option to publish the peer review history of their article (what does this mean?). If published, this will include your full peer review and any attached files.

Reviewer #1: **Yes: **Oscar Mbembela

Reviewer #3: **Yes: **George Kiwango

---

## [Editor Report · Acceptance letter]

28 Sep 2023

PONE-D-23-14105R2 

Thigh-hip ratio is significantly associated with all-cause mortality among Japanese community-dwelling men 

Dear Dr. Kawamoto:

I'm pleased to inform you that your manuscript has been deemed suitable for publication in PLOS ONE. Congratulations! Your manuscript is now with our production department. 

Kind regards, 

on behalf of

Dr Fredirick Lazaro mashili 

Academic Editor

PLOS ONE